# No Object–Location Memory Improvement through Focal Transcranial Direct Current Stimulation over the Right Temporoparietal Cortex

**DOI:** 10.3390/life14050539

**Published:** 2024-04-23

**Authors:** Anna Elisabeth Fromm, Ulrike Grittner, Svenja Brodt, Agnes Flöel, Daria Antonenko

**Affiliations:** 1Department of Neurology, Universitätsmedizin Greifswald, Ferdinand-Sauerbruch-Straße, 17475 Greifswald, Germany; 2Berlin Institute of Health (BIH), 10178 Berlin, Germany; 3Institute of Biometry and Clinical Epidemiology, Charité—Universitätsmedizin Berlin, Corporate Member of Freie Universität Berlin, Humboldt-Universität zu Berlin, and Berlin Institute of Health, 10117 Berlin, Germany; 4Max Planck Institute for Biological Cybernetics, 72076 Tübingen, Germany; 5Institute of Medical Psychology and Behavioral Neurobiology, University of Tübingen, 72076 Tübingen, Germany; 6German Centre for Neurodegenerative Diseases (DZNE) Standort Greifswald, 17489 Greifswald, Germany

**Keywords:** focal transcranial direct current stimulation, current flow modeling, object–location memory, cognitive neuroscience

## Abstract

Remembering objects and their associated location (object–location memory; OLM), is a fundamental cognitive function, mediated by cortical and subcortical brain regions. Previously, the combination of OLM training and transcranial direct current stimulation (tDCS) suggested beneficial effects, but the evidence remains heterogeneous. Here, we applied focal tDCS over the right temporoparietal cortex in 52 participants during a two-day OLM training, with anodal tDCS (2 mA, 20 min) or sham (40 s) on the first day. The focal stimulation did not enhance OLM performance on either training day (stimulation effect: −0.09, 95%CI: [−0.19; 0.02], *p* = 0.08). Higher electric field magnitudes in the target region were not associated with individual performance benefits. Participants with content-related learning strategies showed slightly superior performance compared to participants with position-related strategies. Additionally, training gains were associated with individual verbal learning skills. Consequently, the lack of behavioral benefits through focal tDCS might be due to the involvement of different cognitive processes and brain regions, reflected by participant’s learning strategies. Future studies should evaluate whether other brain regions or memory-relevant networks may be involved in the modulation of object–location associations, investigating other target regions, and further exploring individualized stimulation parameters.

## 1. Introduction

Object–location memory (OLM) is crucial for the adaption to dynamic surroundings throughout life, such as recalling the arrangement of a grocery store to efficiently locate items [1]. OLM functions tend to diminish within normal aging processes as well as in the course of neurodegenerative diseases [2,3,4]. Deficits in OLM, like misplacing objects, can be seen as an early sign of pathological processes in Alzheimer’s disease [2,4]. Quantification of OLM performance in clinical and experimental settings is possible by using neuropsychological tests that assess the capacity to recognize objects, their placements, and later recall them independently [1,5,6]. So far, no effective treatments to encounter deficits in OLM are available [2,3]. Consequently, interventions focusing on improvement or preservation are highly needed, underscoring the importance of studying OLM within the framework of translational research [3].

Successful OLM requires distinct cognitive processes: the identification of an object (object processing), consideration of its location (spatial location processing) and the connection of these information (binding objects to location) [1,7]. These processes involve a large cortical network including fronto-temporo-parietal regions, along with the hippocampus and neighboring structures [7,8,9]. Within the OLM network, object identification is processed by the ventral visual stream, including visual cortices and their projections into the fusiform gyrus and perirhinal cortex within the temporal lobe [7,9]. The processing of the location is carried out by the dorsal visual stream in correspondence to the right parahippocampus and lateral frontoparietal networks [8,9]. Ultimately, the object and its location are connected, requiring medial temporal structures such as the hippocampus and entorhinal cortex [9,10,11]. The OLM network serves as an example for the crucial interplay of cortical and subcortical regions [12,13], with contributions of the angular gyri and the precuneus [7,8,14]. These regions are not only associated with spatial attention [8,15] and long-term storage of mnestic information [16,17], but also involved in short-term memory plasticity processes [18]. OLM training can lead to microstructural plasticity in lateral medial parietal and temporal areas [14,18], causing short-term microstructural and functional alterations in gray matter of the posterior parietal cortex [18]. This suggests that crucial parts of successful OLM are located within these neocortical areas and therefore not only part of the OLM network itself, but also connected within the default mode network (DMN) [19,20]. In the past, the DMN has been targeted successfully by stimulating core nodes of this network with non-invasive brain stimulation [21,22,23,24]. Stimulating parietal areas of the brain with transcranial magnetic stimulation (TMS) led to enhanced memory performances, displaying the possibility to target cortical-hippocampal networks [25]. For OLM, the active stimulation of the cortico-hippocampal network led to recollection precision [26] and might also enhance visuospatial learning [27]. In contrast to TMS, transcranial direct current stimulation (tDCS) is used to influence cognitive functions using subthreshold alteration of the resting membrane potential and increase in excitability in targeted brain regions under the anode [28]. Earlier research has utilized conventional bimodal tDCS using large sponge electrodes [29,30]. This method might not only target a single specific area but also various regions, thereby stimulating different nodes within memory networks [31,32,33]. In the past, stimulating over right (temporo-)parietal brain regions with anodal tDCS have been shown to enhance OLM performance in young [29] as well as in old participants [30] and in patients with prodromal Alzheimer’s disease [2]. However, mixed findings on immediate and delayed performance benefits [30,34,35] as well as negative findings occur [36], displaying a great variability in OLM testing. This variability might not only be influenced by participants’ response to tDCS [37], but their general cognitive ability or their learning strategies [38,39,40,41].

Due to mixed findings and the use of unfocal stimulation, it remains unclear which region should be targeted with tDCS to enhance OLM or whether the activation of multiple regions is the key component that drives the stimulation effect. In the current study, we aimed to examine whether focal tDCS over the right temporoparietal cortex, as a core node of the DMN, would lead to better OLM. To investigate immediate and sustained effects of focal anodal tDCS on OLM performance, we used a focal 4 × 1 montage [32] over the right temporoparietal cortex. We applied current with an intensity of 2 mA and a duration of 20 min (versus 40 s sham) in a between-subject design. During stimulation, participants were administered an OLM task [18]. We hypothesized that the application of 2 mA tDCS over the right temporoparietal cortex during the OLM leads to a superior immediate performance in the anodal compared to the sham group (primary outcome: percentage of correct responses, i.e., remembered stimulus pairs during recall of day 1). The application of tDCS was also expected to lead to larger enhancement of sustained performance in the anodal compared to the sham group, thus, there were sustained effects of the stimulation (day 2). Structural magnetic resonance imaging (MRI) data were collected before the intervention in order to estimate individual stimulation doses. We hypothesized that higher performance benefit would be associated with higher individually induced electric field magnitudes in the target area. As factors like general cognitive ability and learning approaches can influence learning success [38,39,40,41], we additionally aimed to explore their link to training gains. 

In sum, the current study opted to understand whether focal tDCS of the temporoparietal cortex effectively modulates OLM functions and which individual variables at the level of induced current flow, general cognitive ability, and learning strategies determine these effects.

## 2. Materials and Methods

### 2.1. Participants and Study Overview

We conducted a double-blind randomized controlled study comparing the effects of focal anodal tDCS to sham in a between-subject design. Fifty-two healthy right-handed participants (mean age: 22.8 years, SD: 2.6 years, 40 female and 12 male) underwent a baseline neuropsychological testing and MR assessment before a two-day OLM training with 12 h between training sessions (Figure 1A). None of them reported neurological or psychological diseases, severe alcohol consumption, smoking or substance abuse, or psychoactive medication. The study was performed at the University Medicine Greifswald. The study protocol was approved by the ethics committee of the University Medicine Greifswald (BB193/20) and was conducted in accordance with the Helsinki Declaration. Informed consent was obtained before participation. Participants were compensated with 50 EUR. Alternatively, psychology students could receive course credits. The study was preregistered (https://osf.io/h5x6j (accessed on 26 April 2021)).

### 2.2. Baseline Neuropsychological Assessment

At baseline assessment, the cognitive function of each participant was examined with a neuropsychological test battery (Memory: Auditory Verbal Learning and Memory Test [42,43], Rey–Osterrieth Complex Figure Test [44]. Short term/working memory: Digit span [45], Trail Making Test (part A and B; [46]). Executive functions: Stroop (Color-Word-Test) [47], Verbal fluency (Regensburg Verbal Fluency Test) [48]). To examine verbal memory performance, the German version of the Auditory Verbal Learning Test, called “Verbaler Lern- und Merkfähigkeitstest” (VLMT) was used [43]. Percentage of correct responses during the first five consecutive trials was used to operationalize total verbal learning. Visual memory was examined with the Rey–Osterrieth Complex Figure Test (ROCFT) [6,44]. The percentage of correctly drawn components was used to operationalize spatial memory. Participants’ baseline characteristics are summarized in Table 1 with no significant differences between the experimental groups. 

### 2.3. Training Task

Task was reprogrammed using Presentation^®^ software (Version 20.1, Neurobehavioral Systems, Inc., Berkeley, CA, USA, www.neurobs.com). During the training days, participants completed the OLM task on a tablet computer (12.3 inches) that requiring remembering the positions of the second item in pairs of pictures (Figure 1C). Stimuli were everyday items from three different classes: inanimate objects, animals or fruits. Each item was used in two pairs, once as the first and once as the second item. An inanimate object was always paired with two items of the same class (either animals or fruits). For example: the first pair consisted of a ball (inanimate object) and a butterfly (animal), and later a lion (animal) was shown before the ball (inanimate object). Every block consisted of an encoding and recall trial. On the first day of training, three blocks were performed, and the second training started with a recall trial followed by two blocks (Figure 1A). During encoding, participants had to learn the pairing of the unique items on an 8 × 5 grid, therefore 40 associations. The combination of items was randomized for every participant but stayed consistent throughout blocks and training days. At recall, the first part of a pair was shown, and participants had to indicate the location of the second part by touching the screen. To assure the correct understanding of the task and to gain a baseline, participants performed a practice block with different stimuli from the Boston Naming Test [49] on a 4 × 3 grid (12 associations). After training, participants completed a questionnaire regarding potential strategy implementation. They had to decide whether their favored strategy depended on the content or the position of the item. Content-related strategies were operationalized as visualizing the item, inner rehearsal of stimuli’s name and connecting item pairs within sentences. Position-related strategies were operationalized as remembering the position of the item on the whole grid (e.g., on the edge, in a corner) and the relative position of paired items [18]. Memory performance was operationalized by the number of correct responses, thus correctly remembered item positions, during recall. Mean performances per group and recall trial can be found in Appendix A. 

### 2.4. Focal tDCS

Focal stimulation was applied using a 4 × 1 tDCS montage (Figure 1B). The anode was centered over the right temporoparietal cortex, respectively the right angular gyrus (CP6), and was surrounded by four cathodes (FC4, FT8, P2, PO8). Stimulation (2 mA) began with OLM task onset and was delivered for 20 min (fade in/out: 20 s) during the first day of training (Neuroelectrics^®^ Starstim 8, Barcelona, Spain). The return current was equally divided through the four remaining electrodes (−0.5 mA). In the sham stimulation group, the current was ramped up over 20 s and ramped down over 20 s in the beginning of the OLM task (sham in total: 40 s). A conductive gel was injected into the electrode casings. In order to reduce stimulation-induced sensations, a local anesthetic (EMLA Crème^®^ (Aspen Germany GmbH, Munich, Germany), active agent combination of lidocaine and prilocaine) was applied 30 min before the stimulation onset. Adverse events during the stimulation were covered with a questionnaire [50]. Safety outcomes are reported separately as incidences (n, incidence rate with 95%-CI) in total and by anodal stimulation group. Comparisons were performed using incidence rate ratios, based on Poission regression models. Results can be seen in Table 2. 

Participants were randomly assigned to the anodal (n = 26) or sham stimulation group (n = 26) using block wise randomization (https://CRAN.R-project.org/package=blockrand, accessed on 6 April 2020) stratified by age (18–23 years vs. 24–30 years) and sex with a 1:1 ratio. The researcher conducting the experiment was unaware of the group assignment. To assess whether blinding was successful for participants, we asked participants to guess in which stimulation group they were assigned and calculated the James Blinding Index (BI) and the Bang’s BI [51,52]. Participant’s guesses can be seen in Table 3.

### 2.5. MRI Assessment and Electric Field Modeling

MRI acquisition was conducted at the Baltic Imaging Center (Center for Diagnostic Radiology and Neuroradiology, University Medicine Greifswald) on a 3 Tesla scanner (Siemens Verio) using a 32-channel head coil. We acquired 3D T1- (TR = 1690 ms, TE = 2.52 ms, TI = 900 ms, 176 slices, 1.0 × 1.0 × 1.0 mm^3^, flip angle 9°) and T2-weighted (TR = 12770 ms, TE = 86.0 ms, 96 slices, 1.0 × 1.0 × 1.0 mm^3^, flip angle 111°) images. The MRI appointment was not mandatory for participation in the study. We collected data from 47 participants (n = 25 in anodal group). The structural MR scans were used to consider tissue compartments like scalp, skull, cerebrospinal fluid, gray and white matter in order to compute electric field distribution using SimNIBS version 3 [53]. The tissue-specific ohmic conductivities were assigned to the compartments and the modelled stimulation electrodes. The electric field was calculated using numerical methods such as the finite element method [54,55]. The applied stimulation parameters were used to model the electric field: 2 mA at the anode (CP6) and 0.5 mA at each cathode (FC4, FT8, P2, PO8). Electrodes had a diameter of 1 cm and were filled with 3 mm electrode gel. The electric field strength (99th percentile of the magnitude of electric field) during the stimulation |E| in V/m was extracted as mean per participant at the stimulation area.

### 2.6. Statistical Analyses

Linear mixed models (R-software; version 4.1.2) with lme4 und lmerTest [56,57], were conducted for analyzing the memory performance at the OLM task (operationalized by number correct responses, thus correctly remembered item positions, during recall). Each participant was measured over six recalls within two days (Figure 1A). Since we assumed that the stimulation group, recall as well as day have a direct influence on the memory performance, we estimated the effect of each variable as well as their interaction. In order to account for potential confounding factors, the models were adjusted for age and sex. Additionally, the baseline OLM task performance and participant’s strategy (defined by position-related minus content-related strategy) were implemented as fixed factors. Due to technical issues, the value of the baseline OLT of one participant (sham group) was missing and had to be estimated with regression imputation. Participants were included as a random factor to account for random variation in performance (random intercept models). We calculated model-derived marginal means with 95%-confidence interval (CI) for day one and the corresponding last recall to examine immediate tDCS effects on memory performance (primary outcome). The same model with model-derived marginal means for day two and the corresponding last recall was used to examine sustained tDCS effects on memory performance (secondary outcome). Additional information on the linear mixed model can be found in the Appendix A. The influence of electric field strength and neuropsychological data on the overall training gains (recall 1 to 6) was examined via Pearson correlation coefficients (secondary outcomes). Results were deemed as statistically significant if the *p*-value was below 0.05.

## 3. Results

### 3.1. Primary Outcome: Immediate Effects

For the primary outcome (percentage of correct responses on day one) there was no tDCS effect (intervention effect: −0.09, 95%CI: [−0.19, −0.02], *p* = 0.08, model-derived marginal means: 0.42, 95%-CI: [0.33, 0.50] for anodal and 0.51, 95%-CI: [0.43, 0.59] for sham group). Thus, the application of 2 mA tDCS did not lead to superior immediate performance in the anodal stimulation group.

### 3.2. Secondary Outcome: Sustained Effects

No larger enhancement of sustained performance in the anodal stimulation compared to the sham group was found on day two (intervention effect: −0.09, 95%CI: [−0.19, 0.02], *p* = 0.07, model-derived marginal means: 0.65, 95%-CI: [0.57, 0.73] for anodal and 0.74, 95%-CI: [0.66, 0.82] for sham group). Figure 2A shows the task performance over training days in both groups.

### 3.3. Secondary Outcome: Electric Field Strength

Computational modeling confirmed that electric fields were induced in right temporoparietal cortex, and the average distribution of electric field strength is displayed in Figure 2B. We observed no link between electric field strength and task performance (r = −0.09, *p* = 0.654; Figure 2C). Therefore, higher electric field strengths in the anodal stimulation group were not associated with higher training gains.

### 3.4. Secondary Outcome: Link to General Cognitive Ability

Task performance was linked to baseline performance. Participants with higher baseline scores performed better at the training task (β = 0.29, 95-CI: [0.02, 0.55], t_45_ = 2.05, *p* = 0.046). Participants tend to use more content-related (for anodal n = 20, for sham n = 18) than position-related strategies (for anodal n = 6, for sham n = 8). Participants’ learning strategy was associated with task performance; however, it was not statistically significant (β = −0.11, 95%-CI: [−0.217, −0.01], t_45_ = −2.02, *p* = 0.066). Thus, the use of position-related strategies seemed to be less beneficial than the use of content-related strategies.

While baseline spatial memory was not associated with training gains (r = 0.14, *p* = 0.31; Figure 3A), verbal learning showed a weak positive link (r = 0.36, *p* = 0.009; Figure 3B). This indicates superior training gains in participants with higher verbal learning performance. There was no significant influence of the covariates age (β = 0.00, 95-CI: [−0.02, 0.02], *p* = 0.97) and sex (β = 0.04, 95-CI: [−0.07, 0.15], *p* = 0.50) on OLM performance.

### 3.5. Adverse Events and Blinding

Table 2 shows self-reported stimulation sensations. No serious adverse events were reported, and no participant terminated participation due to occurrence of adverse events. In total, 42 participants (anodal n = 23) felt at least mild stimulation sensations. Still, the incidence of adverse events did not differ between groups (IRR = 1.3, 95%-CI: [0.9, 2.0]).

James’ BI estimate was 0.6 (95% CI: 0.5–0.7) implying that participants were effectively blinded [51,52] (Table 3). Bang’s BI for the anodal group was −0.1 (95% CI: −1, 0.2) and 0.1 (95% CI: −1, 0.4) for the sham group. This result indicates that blinding was effective even on closer examination of the two stimulation conditions [51,52].

## 4. Discussion

We examined the effects of focal stimulation over the right temporoparietal cortex on OLM performance. Fifty-two young adults underwent a two-day OLM training with either anodal or sham stimulation administered for 20 min on the first day. Participants were able to learn the task and general effects of training were shown. However, neither immediate nor sustained performance differences were observed between stimulation groups. Baseline performance had a significant influence on training gains, i.e., participants with higher baseline scores performed better at the task. Participants who used content-related learning strategies showed slightly higher training gains compared to participants using position-related learning strategies. There was no link between electric field strength in the anodal stimulation group and training gains, indicating no beneficial effect for individuals with higher field magnitudes in the target region. Spatial memory performance was not associated with training gains at the OLT. Instead, participants with higher scores in verbal learning showed superior training gains.

For the application of tDCS-accompanied OLM training, we are not the first to encounter negative results [36], but rather contributing to the heterogenous findings on memory enhancement. We extend previous findings by considering focal instead of conventional tDCS application. Previously, two different groups investigated the effect of right-hemispheric tDCS on OLM performance, operationalized by allocentric paradigms without spatial navigation parts [29,30]. Beneficial effects on OLM performance for tDCS over the right temporoparietal lobe [29,30,34] as well as mixed findings [35] or negative results have been shown [36]. One paradigm required participants to remember the correct location of different buildings on a street map [2,30,34,35,36]. Immediate effects have been found while the stimulation was applied to patients in prodromal stages of Alzheimer’s disease [2]. The same stimulation can be advantageous for sustained OLM performance in healthy older adults [30,34]. Additionally, this stimulation was linked to beneficial functional connectivity alternations in the DMN [34]. However, the stimulation also resulted in no tDCS-associated improvement at all [36] or only in combination with serotonin reuptake inhibitors [35]. When young healthy adults were tasked with memorizing the positions of cards in a 4 × 4 grid [29], tDCS over the right parietal cortex led to immediate enhanced training gains in the anodal stimulation group.

In our study, the paradigm required encoding both the item positions of everyday objects on a grid and their associations with paired items, which served as recall cues. Unlike previously used tasks, this additional component involving paired items could have prompted greater engagement of higher-order cognitive processes and brain regions besides OLM [1] including associative memory [18]. Moreover, because the study was not conducted as a cross-over design [29], interindividual differences of participants have to be considered carefully. Consequently, several factors might have led to our negative findings, like stimulation-dependent (interindividual variability in responsiveness to tDCS, low induced electric field strength, non-beneficial stimulation location) and paradigm-dependent factors (different approaches of handling the task and cognitive processes involved).

Given that a between-subject design was used, our results are more likely to be influenced by participants’ interindividual variability in responsiveness to tDCS [58,59,60]. This variability is linked to individual aspects of brain anatomy (e.g., gyrification), which has an impact on stimulation effects [61,62]. In particular, focal tDCS compared to bimodal tDCS effects are more likely to be influenced by these interindividual differences, which often are associated with negative results [63,64,65]. For bimodal tDCS with larger electrodes, this influence might be more negligible, as multiple areas rather than precise regions are stimulated [31,32,33]. Due to its impact on widespread areas, it may be easier to target memory networks because multiple nodes might be influenced [66,67], resulting in beneficial tDCS effects for OLM in the past [29,30,34]. When applying focal tDCS, the stimulation location has to be chosen very precisely, since the stimulation radius will be smaller compared to bimodal tDCS and even minor derivations might influence the effect [31,68,69]. Still, it is common to rely on anatomical landmarks or standard systems, like the EEG 10-20 system, rather than on individualized targets [70], although the unique brain response across participants and its influence on stimulation effects have been shown [24]. Recently, there has been growing interest in utilizing TMS in conjunction with neuroimaging techniques to gain insights into brain network interactions [24]. This approach, informed by diffusion-weighted as well as functional MR data, focuses on precise stimulation targeting for enhanced network engagement [23,70]. Studies have successfully improved memory functions by identifying specific stimulation areas, such as the lateral parietal cortex, based on connectivity maps [25]. Targeting the left parieto-hippocampal network through TMS has led to enhanced visuospatial learning and recollection precision, highlighting the effectiveness of targeting nodes within cortico-subcortical networks for enduring changes in memory [26,27]. In the current study, we did not apply individual montages corresponding to the structural or functional MR data of the participant but relied on the standard EEG 10-20 system (anode over CP6). In order to ensure the desired overlap of induced electric field and individual target, the use of the EEG 10-20 system for focal tDCS might not be enough to reach the precise location. Other stimulation studies point to the direction that only the application of individual montages might lead to observable differences at the group level [70,71].

Apart from the necessity of identifying the exact stimulation location for focal tDCS, this approach is also characterized by a lower induced field strength in comparison to bimodal tDCS [31,68]. Usually, the induction of more current in the target region is associated with superior memory performance [72,73]. In order to examine the influence of interindividual differences in the induced current, we estimated the electric field strength during the stimulation as the individual average in the stimulation target [53,74]. Contrary to our expectations, we found no link between electric field strength and training gain. Thus, participants with higher doses of tDCS in the target area did not profit on a behavioral level. This result can either lead to the interpretation that the relationship between performance benefits and field strength is rather complicated and influenced by different anatomical or functional factors [68]. For example, the neurochemistry of the target region during the tDCS application might impact the stimulation effect [75], suggesting a non-linear association between electric field strength and behavioral response [68]. However, our result might also indicate that the induction of more current in the right temporoparietal cortex may not be sufficient to enhance the performance in the anodal stimulation group in this particular task although this target has been beneficial in the past. Therefore, it is conceivable that other regions are more crucially involved in modulating the memory performance of this specific task.

Overall, behavioral findings of tDCS studies tend to show a high inconsistency in response. Missing stimulation effects are often associated with interindividual variability, which is increased both in between-subject designs and when using focal tDCS. We found no association between electric field strength in our target and training gains. It is possible that interindividual differences in the ability to learn or handle the task interfere with a possible stimulation effect because they require additional cognitive processes and therefore underlying brain regions. If areas beyond our target are crucially involved in successful task performance, the focal stimulation over the right temporoparietal cortex might not modulate activity in all or the main involved brain regions.

Stimulation effects, especially in the context of memory enhancement, are influenced by participants’ general cognitive ability and their learning approaches [38,39,40,41,76]. Previously, in healthy young adults with high performance in training tasks, tDCS was not associated with superior OLM training gains [34], similar to other studies in young high performing adults [74]. Consequently, it might be easier to find tDCS effects in groups with lower general performance levels [2,34], which are more challenged by the tasks and at an increasable level [64]. In our study, we observed a significant influence of baseline performance, but no interaction with tDCS. In line with previous findings, the stimulation was ineffective even for a lower performing subsample suggesting no beneficial tDCS effects at all. This implies that some participants are better in learning this task since the beginning and continue to outperform initially low performing participants beyond an influence of the stimulation. This seems to be related to differences in the ability to learn or different approaches of how to handle the learning at this task [76]. In the past, it has been suggested that general cognitive abilities as well as the encoding strategy contribute to OLM functions [76]. In our study, participants had to not only to encode the position of items on a grid [29], but also its association with a paired item, because the paired item served as a cue during the recall. In order to learn these associations, participants can use different strategies: they can focus either on the content of an item (content-related strategy) or on its position (position-related strategy). Those participants who implemented a content-related strategy in contrast to a position-related strategy performed slightly superior. Because the stimuli are everyday objects, they require less early visual processing due to their familiarity and are instead supported by long-term knowledge, suggesting a semantic component [77]. Additionally, the encoding of pictures is linked to a high amount of conceptual processing [78], as they are verbalizable and can be transferred from visual material into a verbal code automatically [79]. For content-related strategies, this phenomenon might be even increased because this strategy contains techniques of inner rehearsal of an item’s name and connecting item pairs within sentences. Moreover, participants reported to connect the different items within a meaningful story (example: the lion kicks the ball). This suggests that participants try to implement the items into pre-existing semantic networks, which is supposed to increase memory performance [39,80]. The influence of learning strategies has been identified as critical contributor to memory performance in the past [39,76]. For example, only participants who implemented a content-related learning strategy, by imagining objects to interact with each other, performed above chance [39]. This indicates that integrating efficient learning strategies might boost memory performance and could be an essential aspect to consider in future tDCS studies targeting memory enhancement. Moreover, data from the neuropsychological baseline supports the link to semantic memory processes, as verbal learning seems to be particularly important for higher training gains. Against the assumption that the task is linked mostly to spatial memory and location processing [1], we found no association of spatial memory and overall training gains. Instead, participants with better total verbal learning showed an increased learning development at the task, suggesting that left-lateralized processes are crucially involved in higher training gains [41]. Therefore, cognitive processes like object processing and binding objects to locations were more involved for successful performance than the mere identification of the position [1]. The role of the left hemisphere for OLM has been discussed in the past, suggesting that both hemispheres are important for successful memory retrieval [8,81]. The semantic component of the task could also explain the additional activation in the lingual gyrus and the left hemisphere in earlier research [18] as these areas are considered to be involved while making logical conjunctions [40].

Due to the semantic aspects of the task, additional relevant networks (like the language network, dorsal attention network or fronto-parietal network) may be activated [18,40], requiring a different stimulation target [21]. For example, language networks can be enhanced by stimulating the left inferior frontal gyrus [41]. Dorsal attention networks seem to have targets at the left superior parietal gyrus [24], whereas fronto-parietal networks have been successfully altered in the past while stimulating the left dorsolateral prefrontal cortex [82]. Within the targeted DMN, TMS studies have shown that the left parietal cortex might be a successful stimulation target to enhance performances in visuospatial learning as well as in recollection precision [25,26,27]. These findings indicate a complex interplay of different cognitive functions, with brain regions across hemispheres contributing to the OLM [8]. Furthermore, it opens the question whether an optimal focalized tDCS that improves performances can only be achieved while considering the nodes of a participant’s individual memory network.

Ultimately, this prompts the inquiry whether a more tailored approach to enhance memory functions with electrical stimulation is needed. Individualized tDCS application has been discussed in recent years, challenging the one-fits-all approach [62,68], for example, by applying individualized doses of tDCS [37]. However, to enhance memory functions, which rely on large networks and various cognitive processes, it might not be enough to only individualize stimulation parameters. As we observed in our data, the interindividual variability in the learning approaches and therefore the underlying processes, might as well contribute to OLM performance. Examining the individual memory network of participants with resting-state functional MRI and targeting the maximally connected cortical nodes with focal stimulation [25] might be a worthwhile approach to engage cortico-hippocampal networks in the future with tDCS.

A limitation of our study is the inclusion of more female than male participants (40 female vs. 12 male). For spatial memory, it is commonly assumed that men excel in mental rotation and spatial orientation tasks, whereas women typically outperform men in OLM tasks [83,84]. Consequently, it is plausible that female and male participants may process the OLM task differently, leading to performance disparities beyond the influence of tDCS [84]. Notably, we considered this issue by ensuring equal distribution of sexes in both the tDCS and sham group (anodal: 20 female/6 male, vs. sham 20 female/6 male), and by incorporating sex as a covariate in the analyses. No significant effect of sex on behavioral performance was observed. Therefore, the influence of sex on OLM performance in this study may be negligible. Nevertheless, future studies should strive for balanced sex distribution to mitigate potential confounding effects. Additionally, the study sample included only young adults (age range: 19–29 years). Due to age-related neural alterations, results cannot necessarily be transferred to older adults [85]. Whether the effects in older samples or in pathology are different has to be investigated in future studies.

## 5. Conclusions

In sum, focal stimulation of the right temporoparietal cortex did not enhance the training gains of the task. This group-level finding was confirmed by non-existent associations of individual benefits with electric field strengths. The association with verbal learning memory suggested that the right-hemispheric modulation of object–location association may be impacted by semantic resources reflected by the participant’s learning strategies. Future studies should evaluate effects of focal tDCS over other targets to investigate whether other networks/regions are involved. Moreover, they should consider investigating the potential of individualized stimulation protocols including the identification of core nodes of memory networks within each participant. Consequently, one possible approach in future research might be the use of neuronavigation and flexible electrode attachments matched to participants’ structural or functional MRI to target the core nodes of memory networks while using focal tDCS.

## Figures and Tables

**Figure 1 life-14-00539-f001:**
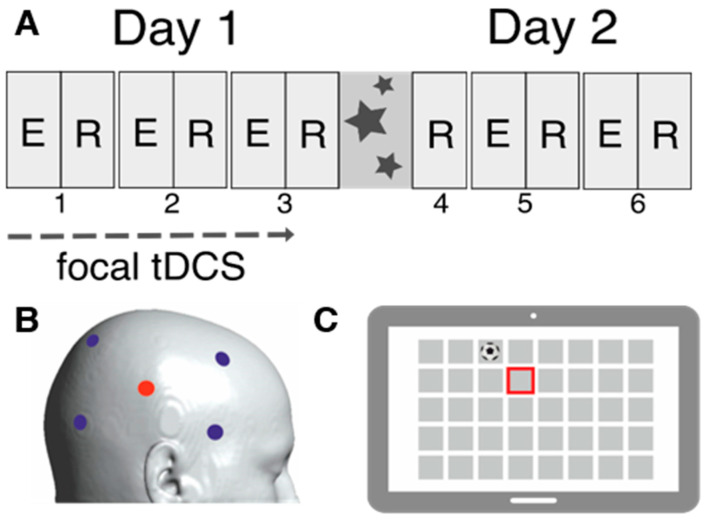
Study design. Summary of study design including procedure of training, stimulation and task description. (**A**): Experimental procedure during the training days. The task contained 6 learning blocks. A learning block is separated into an encoding trial (E) and a recall trial (R). The focal transcranial direct current stimulation (tDCS) was administrated for the first 20 min of the first day. The first day contained three alternating blocks. After an over-night interval (12 h), the second day started with a recall trial followed by two blocks. (**B**): Schematic illustration of the focal tDCS montage. We used the 4 × 1 focal tDCS montage with anode (red) at CP6 and cathodes (blue) at FC4, FT8, P2, PO8. (**C**): Visualization of object–location memory task (OLM). OLM task was completed at a tablet (12.3 inch) with an 8 × 5 grid. At recall, the first item of a pair is presented (e.g., ball), while the position of the paired item has to be selected via touching the screen (red frame).

**Figure 2 life-14-00539-f002:**
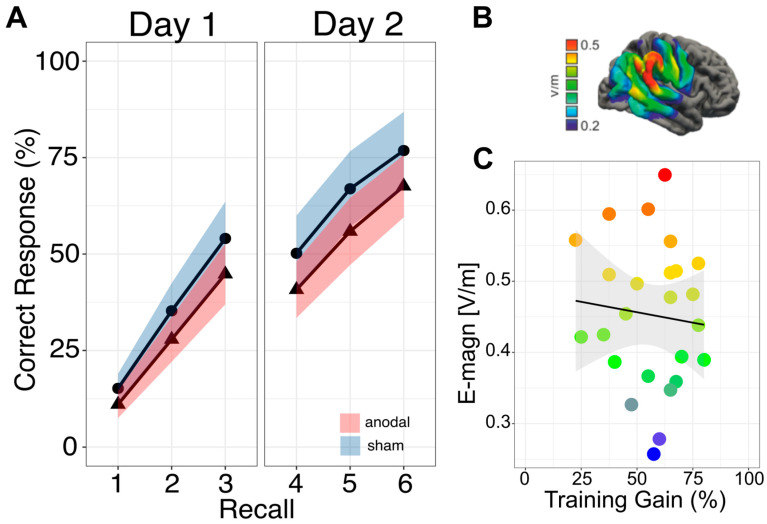
Behavioral and electric field modelling results. (**A**): Visualization of mean performance (in percent) of the stimulation groups during the object–location memory task. Anodal stimulation group as triangle with a red shaded 95% confidence interval. Sham stimulation group as circle with a blue shaded 95% confidence interval. (**B**): Visualization of the electric field strength (99th percentile of the magnitude of electric field) during the stimulation in V/m. The average distribution of electric field strength is displayed from blue to red color (indicating blue areas with low and red areas with high electric field strength). (**C**): Scatterplot to visualize the association between electric field strength (V/m) in the anodal stimulation group (n = 25) and overall training gain (difference recall 6 and recall 1). Colors correspond to the electric field strength from Figure 2B. R = −0.09, *p* = 0.654.

**Figure 3 life-14-00539-f003:**
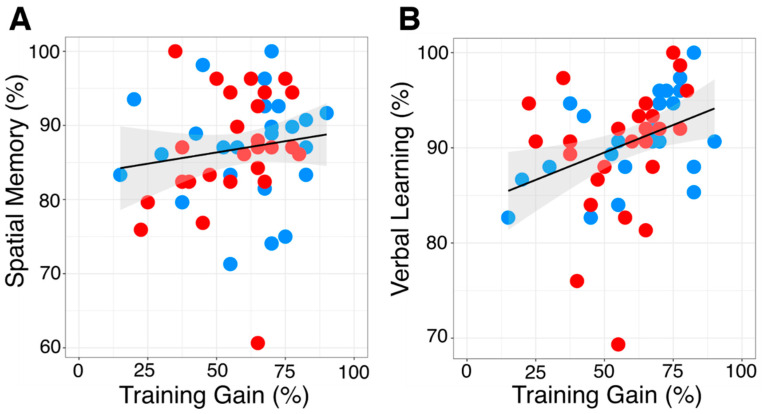
General cognitive ability. Summary of neuropsychological baseline. (**A**): Scatterplot to visualize the association between spatial memory (%) in the anodal (red) and sham (blue) stimulation group (n = 52) and overall training gain (difference recall 6 and recall 1). Shaded areas represent the 95% confidence interval. R = 0.14, *p* = 0.31. (**B**): Scatterplot to visualize the association between verbal learning (%) in the anodal (red) and sham (blue) stimulation group (n = 52) and overall training gain (difference recall 6 and recall 1). Shaded areas represent the 95% confidence interval. R = 0.36, *p* = 0.009.

**Table 1 life-14-00539-t001:** Baseline characteristics.

		Total	Anodal	Sham
N (females)		52 (40)	26 (20)	26 (20)
Age [years]		22.8 (2.6)	22.5 (2.7)	23.2 (2.5)
Verbal Learning [%]		90.1 (5.8)	89.8 (6.8)	91.3 (4.7)
Spatial Memory [%]		86.9 (7.6)	86.7 (8.3)	87.1 (7.0)
Digit Span [n]	forward	8.2 (1.7)	8.2 (1.4)	8.2 (1.9)
	backward	7.6 (1.7)	7.4 (1.5)	7.9 (1.8)
Stroop [s]	words	26.4 (3.2)	26.1 (2.6)	26.7 (3.8)
	color	42.8 (8.3)	42.4 (7.6)	43.2 (9.1)
	interference	63.9 (14.9)	62.5 (11.8)	65.3 (15.9)
TMT [s]	A	19.9 (6.7)	21.1 (7.3)	18.7 (6.0)
	B	40.7 (11.5)	42.0 (13.5)	39.5 (9.2)
Verbal Fluency [n]	phonematic	18.5 (4.3)	18.5 (4.0)	18.6 (4.7)
	semantic	32.9 (6.4)	33.9 (6.7)	31.8 (6.0)
	switch	19.3 (3.0)	19.3 (3.8)	19.3 (2.1)
Baseline OLM [%]		31.5 (16.9)	33.7 (18.6)	29.5 (15.1)

Note. Mean and SD values (except for “N”) are provided. Verbal learning operationalized as correct responses during the first five consecutive trials of the German Auditory Verbal Learning Test (in percent). Spatial memory operationalized as correct responses during the three trials of the Rey–Osterrieth Complex Figure Test (in percent). TMT. Trail Making Test. n. number of correct responses. s. seconds. OLM. Object–location Memory.

**Table 2 life-14-00539-t002:** Self-reported incidence of adverse events (at least mild symptoms) by group during intervention at day one.

	Total n = 52	Anodal Group n = 26	Sham Group n = 26	Incidence Rate Ratio for Group Differences (95% CI)
Total number of adverse events	76	41	35	1.3 (0.9–2.0)
Itching	18/1.6 (0.6)	11/1.8 (0.6)	7/1.1 (0.4)	2.5 (1.1–6.0)
Pain	5/1.4 (0.5)	1/1	4/1.5 (0.6)	0.2 (0.0–1.0)
Burning	18/1.3 (0.6)	13/1.4 (0.7)	5/1 (0)	3.6 (1.4–10.9)
Warmth/heat	17/1.4 (0.6)	9/1.6 (0.7)	8/1.3 (0.5)	1.4 (0.6–3.2)
Metallic taste	1/1	0	1/1	NA
Fatigue/decreased alertness	5/1 (0)	3/1 (0)	2/1 (0)	1.5 (0.2–11.4)
Other	12/1–3 (0.7)	4/1 (0)	8/1.5 (0.8)	0.3 (0.1–1.0)

Note. Reported values are absolute frequencies of the respective adverse events/mean (SD). 1 = mild, 2 = moderate, 3 = strong. Incidence rate (95% CI).

**Table 3 life-14-00539-t003:** Number of participants by group assignment and guess.

Assignment		Response			
		Anodal	Sham	DK	Total
	Anodal	9	12	5	26
	Sham	8	11	7	26
	Total	17	23	12	52

Note. DK denotes “Don’t know”.

## Data Availability

The generated data and the corresponding codes can be found at GitHub (https://github.com/annaelisabethfromm/PlastOLM_statistics (accessed on 23 June 2023)). The (MRI) data are not publicly available due to potential identifying information that could compromise participant’s privacy.

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
