# Peer review of "No Object–Location Memory Improvement through Focal Transcranial Direct Current Stimulation over the Right Temporoparietal Cortex"

_life, 2024, doi:10.3390/life14050539_

Round 1

Reviewer 1 Report

Comments and Suggestions for Authors

The work reported in this manuscript assessed the stimulation of the right temporoparietal cortex on the improvement of object-location memory. The authors reported that stimulation of the right temporoparietal cortex had no improvement effect on object-location memory. Comments and concerns are noted below.

-         The hypothesis of the study was exactly explained but the aim of the study is not clear.

-         According to this sentence "52 healthy right-handed participants (mean age: 22.8 years, SD: 2.6 years, 40 female)". Were the other 12 participants male? How old are they?

-         Why did use only female but not male?

-         The limitation of study should be stated.

Author Response

We thank the reviewer for the valuable comments which we address in detail below. In particular, we improved the phrasing of the manuscript to clarify the aim of the study and avoid misunderstandings with regard the sex distribution in the sample.

Please see the attachment (Reviewer #1).

Reviewer 2 Report

Comments and Suggestions for Authors

The study investigated object location memory (OLM) performance in 52 healthy subjects by applying tDCS over the right temporoparietal cortex. The study provided no enhancement of OLM performance. The authors hypothesized that it might be due to individual learning strategies. Also, the authors provided suggestions to investigate other cortical areas in an individualized manner in future studies. The topic is interesting (however having negative results and is a bit confusing for the reader), however, there are modifications recommended and clarifications to be provided.

Major

The introduction should be updated with more information on other techniques investigating OLM and why OLM investigation is important. Is there any medical (pathologies) background why this knowledge would be beneficial? Are there nTMS studies investigating the same topics, and what are the results?

-Since neuropsychological evaluation was performed for the included subjects, the introduction paragraph is missing a bit more explanation on the neuropsychological background for the OLM performance.

-The last paragraph of the Introduction is a bit methodological description, and therefore it is suggested to rewrite this paragraph and to write the problem and the aim of the study.

-Methodology – why there is no balance in gender, 40 female subjects were included. No significant differences in neuropsychological tests would not be an argument to involve more females in the study. Also, most of the subjects were young/similar age. There is no limitations on this fact in the discussion paragraph.

-Discussion – some clarifications are needed to understand how many groups investigated OLM performance of the right hemisphere with tDCS, and whether the design was similar to that submitted in the manuscript. Also, the discussion section does not bring any previous results on other neuromodulatory techniques such as TMS on this topic, therefore, it is suggested to include these results. The reader might get the impression that this is the first tDCS investigating OLM and having negative results. How the reader could interpret the “importance” of this topic if a negative result is obtained by tDCS, but not bringing some broader explanation on other neuromodulatory techniques (TMS) results?

Minor

-Raw 430 – the sentence should be removed.

Author Response

We thank the reviewer for the valuable comments and suggestions. In particular, we are confident that we were able to improve the introduction by extending the neuropsychological background of OLM as well as its clinical relevance. We updated our manuscript by incorporating results from TMS studies to improve the comprehensibility of our findings. Our detailed responses are listed below.

Please see attachment (Reviewer #2)

Reviewer 3 Report

Comments and Suggestions for Authors

The paper titled "No object-location memory improvement through focal transcranial direct current stimulation over the right temporoparietal cortex" by Fromm et al. explores the impact of transcranial direct current stimulation (tDCS) on object-location memory (OLM). The study involved 52 participants who underwent a two-day OLM training with either anodal tDCS or sham stimulation over the right temporoparietal cortex. The results indicated that focal stimulation did not enhance OLM performance on either training day. Interestingly, participants with content-related learning strategies showed slightly better performance, and training gains were associated with individual verbal learning skills. The study highlighted the complexity of memory functions and suggested that different cognitive processes and brain regions might be involved in OLM enhancement. Future research directions could include exploring other brain regions or memory-relevant networks for modulation and investigating individualized stimulation parameters to improve memory performance.

In general, I think the idea of this article is really interesting and the authors’ fascinating observations on this timely topic may be of interest to the readers of Life. However, some comments, as well as some crucial evidence that should be included to support the author’s argumentation, needed to be addressed to improve the quality of the manuscript, its adequacy, and its readability prior to the publication in the present form. My overall judgment is to publish this paper after the authors have carefully considered my suggestions below, in particular reshaping parts of the ‘Introduction’ and ‘Methods’ sections by adding more evidence.

Please consider the following comments:

A graphical abstract that will visually summarize the main findings of the manuscript is highly recommended.

In general, I recommend authors to use more evidence to back their claims, especially in the Introduction of the manuscript, which I believe is currently lacking. Thus, I recommend the authors to attempt to deepen the subject of their manuscript, as the bibliography is too concise: nonetheless, in my opinion, less than 80 articles for a research article and a meta-analysis are really insufficient. Therefore, I suggest the authors to focus their efforts on researching more relevant literature: I believe that adding more studies and reviews will help them to provide better and more accurate background to this study.

While the introduction provides a comprehensive overview of object-location memory (OLM) and the potential benefits of transcranial direct current stimulation (tDCS), it would be beneficial to expand on the neural substrates involved in OLM. Specifically, I would ask the authors to add details about the specific neural networks, to enhance the understanding of how these brain areas contribute to spatial cognition and memory processes [1-2]. By elaborating on the neural underpinnings of OLM and how they interact within broader cognitive networks, the paper can offer a more nuanced perspective on the mechanisms underlying object-location memory and its modulation through tDCS.

In my opinion, the study did not show significant immediate or sustained performance differences between the anodal and sham stimulation groups. Therefore, I would ask the authors to further discuss the implications of these findings and whether the stimulation parameters used were optimal for inducing cognitive enhancements.

The paper mentions that participants with content-related learning strategies showed slightly superior performance. I believe that it might be valuable to delve deeper into how different learning strategies influenced the outcomes and whether adjusting the training approach could yield more significant results.

The authors reported that they have collected data on individual electric field magnitudes; however, I believe that it would be pertinent to analyze how these magnitudes correlated with performance benefits. Could they provide more statistical details and further explore this relationship, to provide insights into the effectiveness of the stimulation technique used in the study?

Given the complexity of memory functions and brain networks, I would suggest to better discuss whether targeting other brain regions or memory-relevant networks could potentially lead to different outcomes. This could open up avenues for future research to explore alternative stimulation targets for enhancing object-location memory.

Considering that factors like general cognitive ability and learning approaches can impact learning success, a discussion on individualized stimulation parameters tailored to participants' cognitive profiles could be beneficial. This personalized approach may enhance the efficacy of transcranial direct current stimulation in improving memory performance.

I hope that, after careful revisions, the manuscript can meet the journal’s high standards for publication. I declare no conflict of interest regarding this manuscript.

Best regards,

Reviewer

References:

1. https://doi.org/10.3390/ijms25020864

2. https://doi.org/10.3390/ijms25052724

Comments on the Quality of English Language

Minor english check is required.

Author Response

We thank you for the general appreciation of our manuscript and have thoroughly addressed your concerns to improve the quality of the manuscript, its adequacy, and its readability. As recommended, we extended the theoretical background of the introduction, clarified the potential link between electric fields and memory performance, and deepened our discussion on how different learning strategies can affect memory performances.

Please see attachment (Reviewer #3)

Round 2

Reviewer 2 Report

Comments and Suggestions for Authors

The authors answered to the raised comments. Thank you.

Reviewer 3 Report

Comments and Suggestions for Authors

Dear Authors,

I am pleased to acknowledge that you have indeed addressed all of my concerns and queries in a clear and precise manner. Your responses have provided valuable insights into the modifications made to the manuscript in light of my comments. It is evident that you have taken great care to ensure that the revised manuscript aligns more closely with the scientific rigor expected for publication in Life. Having reviewed the revised manuscript, I am satisfied with the changes that have been implemented.

In light of the above, I am pleased to recommend acceptance of your manuscript for publication in Life.

Best regards,

Reviewer